# Towards causal prediction on Magnetic Resonance Imaging including non-imaging data

**Louisa Fay**[1,2]                                    louisa.fay@med.uni-tuebingen.de
**Florian Marencke**[2]                                florian.marencke@posteo.de
**Bin Yang** [2]                                       bin.yang@iss.uni-stuttgart.de
**Sergios Gatidis**[3]                                 sgatidis@stanford.edu
**Thomas Kuestner**[1]                                 thomas.kuestner@med.uni-tuebingen.de

[1] *Medical Image And Data Analysis (MIDAS.lab), University Hospital of Tuebingen, Germany*

[2] *University of Stuttgart, Stuttgart, Germany*

[3] *Stanford University, Department of Radiology, Stanford, USA*

**Editors:** Under Review for MIDL 2024

## Abstract

Deep learning methods can detect correlations in data but they cannot determine underlying causal relationships. Understanding causality, however, is essential because spurious correlations can obscure the true relationships in the data. In many large studies, imaging data is accompanied by additional tabular (non-imaging) clinical data. Our aim is to use the non-imaging information to learn a multi-modal feature representation that can make predictions based on learned causal dependencies while avoiding spurious correlations. This work presents our first preliminary results and outlines our future investigations.

**Keywords:** Causality, MRI, deep learning, multi-modal pre-training, tabular data

## 1. Introduction

Deep learning (DL) has the potential to fundamentally transform clinical workflows. However, DL methods can only detect correlations in data but are incapable of determining their causal meaning.(Marcus, 2018) Relying solely on learned correlations could lead to incorrect or biased conclusions, as these models do not capture the true causal relationship between input and output.(Veitch et al., 2021) In medical imaging, various factors, such as the type of scanner or acquisition conditions, can influence the acquired image. Therefore, DL models trained on this data tend to learn spurious correlations instead of task-specific features. Since many medical imaging databases are accompanied by non-imaging clinical tabular information, we attempt to leverage the additional non-imaging data to learn a feature space that reduces the influence of spurious correlations and enforces predictions based on causal relationships. Hence, by effectively utilizing non-imaging information, we aim to improve the accuracy and reliability of DL models in medical imaging.

Our methodology adapts the previous idea of CLIP (Contrastive Language-Image Pre-Training) (Radford et al., 2021), which learns a multi-modal feature space based on self-supervised contrastive learning between image-text pairs. The CLIP model uses image captions as text modality. Our idea is to automatically generate causal directed acyclic graphs (DAG) between the available tabular features to enrich the feature space not only

with tabular information but also with information about causal relationships. By differentiating between causal features and spuriously correlated feature-based information gained by the causal DAG - the aim is to create a new contrastive loss-term for the pre-training of the feature space that pulls causally related non-imaging features to the imaging features while pushing spuriously correlated non-imaging features apart.

In this work, we present our preliminary results, which form the foundation to achieve our outlined objectives. These preliminary results include the following novel contributions: (1) Training a multi-modal feature space based on the Alzheimer's Disease Neuroimaging Initiative (ADNI) (ADN) database, comprising of 3,402 brain MRI and non-imaging tabular data. In this preliminary work, the tabular data is not yet represented as causal DAG, but a novel approach is applied by transforming the tabular data into a natural language string. (2) Pre-training the feature space on three different sets of tabular features which demonstrates performance improvements on uni-modal downstream task prediction using the pre-trained multi-modal feature space. (3) Demonstrating first steps towards reduction of spuriously correlated predictions by randomly selecting non-imaging features.

## 2. Methods

**Model:** Our proposed multi-modal pre-training model (Fig.1A) is based on an image encoder and a text encoder. The image encoder is a ResNet-50 that encodes the brain MRI volume into an image embedding vector $x_i \in \mathcal{R}^{256}$. The text encoder consists of a pre-trained DistilBERT tokenizer and model. (Sanh et al., 2019) Input to the text encoder is a string which is generated from tabular data containing each feature as *column name: cell value* and encoded into a vector of $t_i \in \mathcal{R}^{256}$. Subsequently, the similarity between all $N$ embedding vectors $X = x_1, ..., x_N$ and $T = t_1, ..., t_N$ is computed using the cross entropy loss. During pre-training, the model with the highest mean accuracy of sex and Alzheimer's Disease (AD) classification is saved. This model is used for the initialization of the image encoder in the downstream task. Fine-tuning for uni-modal image classification is performed in a supervised manner. Thus, the final layer of the pre-trained image encoder is switched to a classification head sized according to the number of classes of the downstream task.

**Settings:** All pre-trainings are performed three times for 40 epochs (batch size: 64) using ADAMW optimizer with a learning rate of $1.77 \times 10^{-4}$ and $5.68 \times 10^{-5}$, respectively.

**Data:** All used data belongs to the ADNI database (ADN). Pre-training is performed on 4536 samples (training: 3402, validation: 1134) (Fig.1B). Downstream tasks are tested on 1134 additional samples. Each sample includes both modalities, the image (T1-weighted 3D brain MRI with resolution $1 \times 1 \times 1mm^3$) and 29 tabular features.

**Experiments:** Pre-trainings are carried out by using: (1) All 29 tabular features, (2) all tabular features excluding downstream task labels, or (3) a set of 14 randomly selected features. The downstream task predictions are uni-modal using only the image encoder to predict either sex (male/female) or prevalence of AD (cognitive normal; CN, mild cognitive impairment; MCI or AD) from brain MRI. Downstream tasks are evaluated for zero-shot prediction and after fine-tuning for 1, 5, and 20 epochs. Comparisons were made to a reference model with the same architecture as the image encoder initialized on pre-trained ImageNet weights.

## 3. Results and Discussion

Our proposed model achieved for both downstream tasks higher accuracy on the test dataset than the ImageNet-pretrained reference model (Fig.1C). When applying only a selected set of tabular features, zero-shot sex prediction achieved 96% accuracy, which the reference model only achieves after 20 epochs of fine-tuning. We opted to further sharpen the focus of the tabular information by selecting random features for pre-training. A selected feature set with 14 features performed best. Interestingly, the AD prediction performance increased, although age, which is known as a primary risk factor for AD, was excluded from the feature set. This might indicate that the inclusion of the feature age in the other two settings may have affected the pre-trained feature space by learning a spurious correlation between AD and age instead of learning a representation based on causally linked features. However, one of the selected features was APOE4, a genetic risk factor for AD. This information can causally enrich the imaging data and improve the accuracy of AD prediction.

However, we acknowledge several limitations of this preliminary work. In particular, the conclusions drawn need to be further validated. Thus, we will include other databases (UK Biobank, German National Cohort) and compare the results with other methods (Hager et al., 2023). In addition, we observed that more fine-tuning epochs improve the performance of the reference model, bringing it closer to the performance of our best model. This may be attributed to the well-known problem of catastrophic forgetting of pre-trained features, which we will address by extending the learning process with a weight-ensembling method (Marsden et al., 2024). Moreover, we will focus on incorporating causal dependencies into the multi-modal pre-training to control the influence of learned spurious correlations. In conclusion, we showed an efficient way to enhance downstream tasks performance with multi-modal information. We leveraged self-supervised pre-training to take advantage of tabular data that is merged with imaging data.

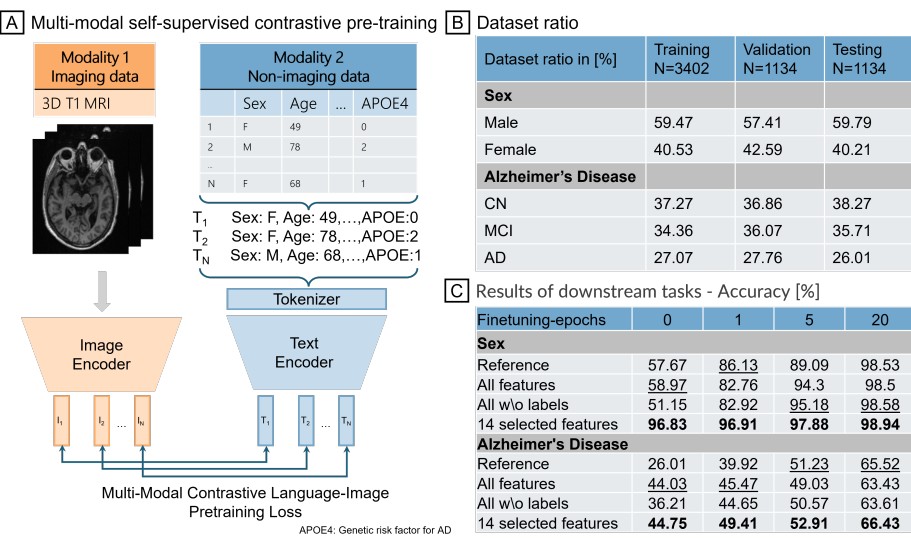

Figure 1: (A) Our multi-modal pre-training model, (B) dataset information, (C) the results.

## Acknowledgments

Data collection and sharing for this project was funded by the Alzheimer's Disease Neuroimaging Initiative (ADNI) (National Institutes of Health Grant U01 AG024904) and DOD ADNI (Department of Defense award number W81XWH-12-2-0012).

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
