# OpenReview forum: "Towards causal prediction on Magnetic Resonance Imaging including non-imaging data"
_MIDL.io/2024/Short_Papers — MIDL 2024 Short Papers_

### Official Review · Reviewer_onvU · 2024-04-24

**Confidence:** 5
**Final Rating:** 3.5

**Review:**

This paper presents a multimodal (imaging and non-imaging biomarkers) self-supervised contrastive pre-training for understanding the causal relationship in AD/MCI diagnosis. The imaging data is fed into the model through the image encoder, while the non-imaging data goes through the text encoder. While the method technically sounds good, it has not been compared with any existing methods.

---

### Decision · Program_Chairs · 2024-04-26

Accept